# Volumetric Spore Traps Are a Viable Alternative Tool for Estimating *Heterobasidion* Infection Risk

**László Benedek Dálya [1], Miloň Dvořák [1,\*] and Petr Sedlák [2]**

[1] Department of Forest Protection and Wildlife Management, Mendel University in Brno, Zemědělská 3, 61300 Brno, Czech Republic
[2] Paměť krajiny, s.r.o., Všetičkova 5, 60200 Brno, Czech Republic
**\*** Correspondence: milon.dvorak@mendelu.cz

**Abstract:** Prophylactic stump treatments against the conifer root rot fungus *Heterobasidion annosum* s.l. should take into account the airborne inoculum density of the pathogen, in order to be economically feasible. Our objective was to test if an automatic volumetric spore trap (AVST) could be used as a sensitive alternative of passive traps for quantifying *Heterobasidion* airspora in forest stands. The routinely used wood disc exposure method (WDE) was implemented simultaneously with AVST in five coniferous monocultures and three near-natural stands without silvicultural management in Czechia. Air sampling took place for 24-h intervals in five months of the main sporulation period. The atmospheric concentration of *H. annosum* s.l. spores ($c_{Ha}$) was determined by qPCR with generic and species-specific primers. AVST detected more target species than WDE on 77% of sampling occasions. When comparing the relative abundance of the three European *H. annosum* s.l. species detected by AVST and WDE, *H. parviporum* and *H. abietinum* were found to be overrepresented on wood discs. $c_{Ha}$ in November was significantly higher than in May and June, confirming the seasonal pattern typical for temperate Europe. For an accurate and rapid estimation of *Heterobasidion* primary infection risk, the usage of AVST is highly recommended.

**Keywords:** aerobiology; conifer root rot; *Heterobasidion*; qPCR detection; spore dispersal; spore trap

## 1. Introduction

The white-rot basidiomycete *Heterobasidion annosum* sensu lato causes extensive damage to boreal and temperate forests, especially conifer plantations [1]. Healthy forest stands are infected by airborne basidiospores of the pathogen, which colonize freshly exposed wood surfaces [2]. Asexual spores (conidia) of *Heterobasidion* are produced in small amounts in nature and are deemed important only in short-distance transmission [2]. Primary infections typically occur immediately after the first thinning, as the residual stumps are a favorable substrate for the germination and establishment of the fungus. Preventive stump treatments are most efficient and economical when applied depending on inoculum availability [3]. Therefore, it is of interest to investigate techniques that could predict the local infection risk by providing accurate data on the spore load of *Heterobasidion* in the air.

Conifer stem discs or culture media have been used as passive spore traps to study the spatial and temporal patterns of *Heterobasidion* spore dispersal for several decades in Europe and North America [4,5]. A major drawback of these methods is their time-consuming and labor-intensive nature. Suction traps designed by Hirst [6], also known as Burkard or volumetric spore traps, have been implemented in a number of aerobiological studies, including those on forest pathogens such as *Ganoderma* spp. [7], *Erysiphe alphitoides* and *Armillaria* spp. [8], as well as serious invasive pathogens, such as *Dothistroma septosporum* [9], *Melampsora larici-populina* [8] and *Hymenoscyphus fraxineus* [10–12]. These constructions are suitable for trapping particles larger than 2 μm [13], such as the basidiospores and conidia of *H. annosum* s.l., both of which are ca. 3–6 μm in diameter [1,14]. To

our knowledge, volumetric samplers for the detection of *Heterobasidion* airspora under forest conditions have been used in an experiment [15], where species-level identification was not attempted. Furthermore, the inoculum of *Heterobasidion* genus has been detected in a proof-of-concept experiment in microscopic slides provided by the French aerobiological network, which uses the volumetric spore traps for allergen monitoring [8].

The purpose of our study was to test the usefulness of an automatic volumetric spore trap (AVST) for quantifying the airspora of all three European *Heterobasidion* species. Our protocol included DNA extraction from the air samples, followed by qPCR. Simultaneously, the wood disc exposure method (WDE) was employed as a well-established way of measuring the spore deposition of *Heterobasidion* [16]. We hypothesized that AVST is more sensitive than WDE—capable of detecting lower inoculum densities of each target species (with an arbitrary sampling effort for conidiophores on wood discs). The specific aims of this work were (i) to compare the performance of AVST and WDE; (ii) to estimate the risk of airborne infection of stumps by *H. annosum* s.l. in different forest types, based on density of inoculum; and (iii) to determine seasonal patterns of spore dispersal in Central Europe.

## 2. Materials and Methods

### 2.1. Air Sampling

Eight locations were selected for the experiment in SE Czechia (Table 1). Five stands were coniferous monocultures (tree species see in Table 1), where the presence of *Heterobasidion* fruiting bodies within 50–100 m from the installed spore traps was confirmed by a visual inspection. According to the model of Stenlid [17], only 0.1% of spores travel further than 100 m from the fruiting body, so the probability of detection would considerably decrease beyond this distance. The other three sites were uneven-aged broadleaved or mixed forests without any symptoms of *Heterobasidion* infection. These near-natural stands were located within nature reserves where forest management ceased more than a century ago. Air sampling took place for 24-h intervals monthly from mid-April until early November 2018, except the hottest summer months (July–August), i.e., five samplings were performed at each site.

**Table 1.** Main features of experimental sites.

| Site | Latitude N | Longitude E | Stand Age (Years) | Dominant Tree Species | Management | *Heterobasidion* Species Present [1] |
|------|-----------|-------------|-------------------|----------------------|------------|--------------------------------|
| 1 | 49.300284 | 16.601441 | 12 | *Picea abies* | managed | *abietinum* |
| 2 | 49.269397 | 16.684964 | 45 | *Picea abies* | managed | *annosum* |
| 3 | 49.325002 | 16.719278 | 107 | *Picea abies* | managed | *annosum* |
| 4 | 49.070277 | 15.674212 | 62 | *Picea abies* | managed | *annosum* |
| 5 | 48.912336 | 17.158900 | 71 | *Pinus sylvestris* | managed | *annosum* |
| 6 | 49.222491 | 16.673499 | 0–149 | *Quercus petraea* | unmanaged | - |
| 7 | 49.283868 | 16.734009 | 0–49 | *Fagus sylvatica* | unmanaged | - |
| 8 | 49.304245 | 16.696342 | 0–146 | mixed [2] | unmanaged | - |

[1] Basidiome within 50–100 m from the installed spore traps. [2] *Fagus sylvatica*, *Carpinus betulus*, *Picea abies*, *Abies alba*.

Two AVSTs made by AMET (Velké Bílovice, Czechia) were put in operation as described previously [9]. Exposed melinex tapes were cut lengthwise into two halves and stored in 2 mL microtubes at −20 °C.

Simultaneously, spores of *Heterobasidion* were captured using WDE [18], modified according to Gonthier et al. [16]. Wood discs, 7–8 cm diameter, cut from *Picea abies* trees were employed. At each sampling site, four open Petri plates were placed 5 m from the AVST along the four cardinal directions. One closed Petri plate served as control. After exposure for 24 h, discs were sprayed with a benomyl solution (10 mg benomyl, 0.5 L ethanol, 1 L distilled water) and incubated at 25 °C. Emerging colonies of the conidial

stage of *Heterobasidion* were counted microscopically at weekly intervals for 3 weeks after trap exposure. We assumed that each colony results from deposition of one viable spore. Based on the area of each disc of ca. 0.0044 m², spore deposition values were recalculated as the number of viable spores·m$^{-2}$·h$^{-1}$ (deposition rate, DR) [19].

### 2.2. Identification of Heterobasidion annosum s.l.

The PCR identification of *H. annosum* s.l. basidiomes found in sites 1–5 (Table 1) was undertaken as described earlier [20], except that the recommended thermal cycling protocol for the Phire Plant Direct PCR Kit (Thermo Scientific) was followed.

From the wood discs, up to eight conidiophores per sampling site were isolated each month except April under a dissecting microscope (magnification 90×), followed by amplification of the translation elongation factor 1-alpha gene [21] by the Phire Plant Direct PCR Kit, using the dilution protocol with 2 μL supernatant as template. PCR conditions and the subsequent identification process were the same as for basidiomes.

### 2.3. DNA Extraction from Air Samples and qPCR

DNA was extracted from air samples, including an empty microtube as a negative control for each extraction run. Spores were disrupted as described previously [10] and processed with the DNeasy Plant Mini Kit (QIAGEN). DNA was eluted twice with 50 μL Buffer AE.

qPCR was carried out on a LightCycler 480 Instrument II (Roche Diagnostics, Basel, Switzerland) in accordance with the manufacturer's instructions. Each sample was amplified with primers matching all three *H. annosum* s.l. species of interest [22], and also with species-specific primers in a duplex assay for *Heterobasidion abietinum/Heterobasidion annosum* s.s. and in singleplex for *Heterobasidion parviporum* [23]. Each 20 μL reaction contained 10 μL LightCycler 480 Probes Master (Roche Diagnostics Netherlands BV, Almere, the Netherlands), 7.5 μL DNA, primers with a final concentration of 0.5 or 0.3 μM, and probes with a final concentration of 0.25 or 0.1 μM for the generic and species-specific assays, respectively. Reactions were performed in triplicates including a no-template control.

### 2.4. Absolute Quantification

Basidiospore suspensions from mature fruiting bodies of all three European *H. annosum* s.l. species were prepared in Nonidet P40-substitute (AppliChem, Darmstadt, Germany) by means of a Bürker chamber. Serial dilutions of $1 \times 10^2$ to $1 \times 10^6$ basidiospores in 250 μL 0.1% Nonidet were pipetted into 2 mL microtubes with melinex tape coated by medical vaseline to mimic the content of air samples. Further processing followed the protocol for the AVST samples. The absolute quantification (fit points method) of *H. annosum* s.l. DNA in the air samples was performed using standard curves of serial dilutions of spore suspensions generated by the LightCycler 480 Software v. 1.5 (Roche). Color compensation for the dye combination FAM-ROX-Cy5 was applied on all runs of species-specific qPCR assays. The atmospheric concentration of *H. annosum* s.l. spores was calculated according to Equation (1):

$$c_{Ha} = \frac{2 \times x_{Ha}}{I \times D \times 0.001} \tag{1}$$

where: $c_{Ha}$ = atmospheric concentration of *H. annosum* s.l. spores (spores·m$^{-3}$); $x_{Ha}$ = number of *H. annosum* s.l. spores detected in the sample; I = intensity of sampling (L·min$^{-1}$); and D = duration of sampling (min). The intensities measured by an electronic anemometer (Trotec TA300, Trotec, Heinsberg, Germany) were $I_1 = 14.04$ L·min$^{-1}$, $I_2 = 19.21$ L·min$^{-1}$.

The limits of detection (LOD) and quantification (LOQ) were interpreted and determined as suggested by Klymus et al. [24]. For each assay, the LOD was determined as the lowest standard concentration with 100% detection. Additionally, the LOD and LOQ were calculated for each *H. annosum* s.l. species using the R script developed by Klymus et al. [24]. The LOQ was defined as the lowest standard concentration with a coefficient of variation

below 35%. Detections below LOD were reported as positive [24], given that at least two replicates had been properly quantified (not extrapolated from the standard curve) at $C_q <$ 40.

### 2.5. Statistical Analyses

For the left-censored variable $c_{Ha}$, values lower than the LOD were imputed from a continuous uniform distribution [25]. Exploratory data analysis had shown that DR and $c_{Ha}$ could be best approximated by a Gamma and Weibull distribution, respectively. Therefore, a rank correlation analysis with Kendall's τ was applied to these variables. Comparison of *H. annosum* s.l. spore abundance in managed and unmanaged stands was undertaken using the Mann–Whitney U test. The effect of seasonality on *Heterobasidion* spore dispersal was analyzed using the Kruskal–Wallis test with post hoc Dunn's test of multiple comparisons and Bonferroni adjustment.

Daily values of mean air temperature (T), relative humidity (RH), total rainfall (RF), and average wind speed (F) were obtained from the closest meteorological station of the Czech Hydrometeorological Institute. These climatic variables are known to influence spore production and deposition for *Heterobasidion* spp. [26]. The relationships between meteorological factors and spore densities were analyzed by rank correlation with Kendall's τ. Average values of T, RH, and F, and cumulative values of RF over the 5-day and 2-week periods preceding sample collection dates were used.

The level of significance was $p < 0.05$ for all tests. Statistical analyses were performed in RStudio [27].

## 3. Results

### 3.1. Performance of qPCR Assays

When determined as a discrete threshold, the LOD was 1000 spores per air sample for all tested *H. annosum* s.l. species (Table 2). Using curve-fitting modeling, the LOD was lower by nearly 50%. As single-copy genes were targeted, these numbers correspond to 75 and 45 nuclei per PCR tube assuming 100% DNA extraction efficiency. Thus, the performance values of the species-specific qPCR assays were the same order of magnitude as determined during assay development. The LOQ was four to nine times higher than the LOD. In the case of the generic assay, LOD and LOQ equaled 100 spores per sample (Table 2).

**Table 2.** Limits of detection (LOD) and quantification (LOQ) in qPCR assays. HA, *Heterobasidion annosum* s.l.; Ab, *Heterobasidion abietinum*; An, *Heterobasidion annosum* s.s.; Pa, *Heterobasidion parviporum*; nd, not determined. Bold values were used for the interpretation of qPCR data.

| Target | LOD in Ioos et al. [23], Nuclei/Rxn | LOD as a Discrete Threshold | | LOD with Curve-Fitting Modeling | | LOQ with Curve-Fitting Modeling | |
|---|---|---|---|---|---|---|---|
| | | Spores/Sample | Nuclei/Rxn | Spores/Sample | Nuclei/Rxn | Spores/Sample | Nuclei/Rxn |
| HA | nd | **100** * | ≤7.5 | nd | nd | nd | nd |
| Ab | ~30 | 1000 | ≤75 | **589** | ≤45 | **5460** | ≤410 |
| An | 311 | 1000 | ≤75 | **589** | ≤45 | **3297** | ≤248 |
| Pa | 29 | 1000 | ≤75 | **589** | ≤45 | **2368** | ≤178 |

* Equal to LOQ.

### 3.2. Comparison of Spore Trapping Methods

The DNA of *H. annosum* s.l. was present in all air samples (Table 3), not reaching LOD in 15% of samples (Supplementary Data, Table S1). Contrarily, *Heterobasidion* colonies developed on at least one wood disc on 50% of sampling occasions (Table 3; Supplementary Data, Table S2). No significant correlation was found between DR and $c_{Ha}$. The composition of air samples was most frequently dominated by *H. annosum* s.s., while *H. abietinum*, and especially *H. parviporum* were prevalent among colonies on wood discs (Table 3).

**Table 3.** Comparison of the suitability of automatic volumetric spore trap (AVST) and wood disc exposure method (WDE) for the detection of airspora composition of *H. annosum* s.l. −, target not detected; o, *H. annosum* s.l. detected but not identified at the species level; +, target present in lower concentration than other target species; ++, target had the highest concentration among all target species; R, result of the comparison; nd, not determined because identification at the species level was not attempted in the case of WDE; A, AVST was more sensitive; W, WDE was more sensitive; I, inconsistent species composition.

| Site | April AVST Ab | An | Pa | WDE Ab | An | Pa | R | May AVST Ab | An | Pa | WDE Ab | An | Pa | R | June AVST Ab | An | Pa | WDE Ab | An | Pa | R | September AVST Ab | An | Pa | WDE Ab | An | Pa | R | November AVST Ab | An | Pa | WDE Ab | An | Pa | R |
|---|---|---|---|---|---|---|---|---|---|---|---|---|---|---|---|---|---|---|---|---|---|---|---|---|---|---|---|---|---|---|---|---|---|---|---|
| 1 | + | ++ | − | − | − | − | A | o | o | o | − | − | − | A | ++ | + | + | − | − | − | A | ++ | ++ | + | o | o | o | A | o | o | o | ++ | − | − | W |
| 2 | + | ++ | ++ | o | o | o | nd | − | ++ | − | − | − | ++ | I | − | ++ | − | − | − | − | A | ++ | + | − | − | − | ++ | I | − | ++ | + | ++ | − | − | I |
| 3 | + | ++ | + | − | − | − | A | ++ | + | ++ | − | − | ++ | A | + | + | ++ | − | − | − | A | ++ | + | + | + | − | ++ | A | + | ++ | ++ | ++ | + | − | A |
| 4 | − | ++ | − | o | o | o | nd | ++ | ++ | ++ | − | − | − | A | − | ++ | − | − | − | − | A | − | + | ++ | − | − | − | A | − | ++ | − | o | o | o | A |
| 5 | − | ++ | − | o | o | o | nd | − | ++ | − | − | + | ++ | W | − | ++ | − | − | − | − | A | + | ++ | − | − | − | − | A | − | ++ | − | ++ | − | − | I |
| 6 | o | o | o | o | o | o | nd | − | ++ | − | − | − | ++ | I | − | − | ++ | − | − | − | A | ++ | − | ++ | − | − | − | A | − | ++ | − | − | − | − | A |
| 7 | − | ++ | − | o | o | o | nd | ++ | − | − | − | − | ++ | I | o | o | o | − | − | − | A | ++ | − | ++ | − | − | − | A | − | ++ | − | − | − | − | A |
| 8 | ++ | + | ++ | − | − | − | A | + | − | ++ | − | − | ++ | A | ++ | + | ++ | − | − | − | A | + | + | ++ | ++ | − | + | A | + | + | ++ | − | − | − | A |

The airspora composition detected by AVST and WDE differed in all cases where a comparison could be made (Table 3). AVST was able to detect more target species 77% of the time, whereas WDE was more sensitive in this regard in 6% of cases. For the remaining sampling occasions (17%), the compositions of target species determined by both methods were fully inconsistent.

### 3.3. Effects of Site, Season, and Weather on the Abundance of Heterobasidion annosum s.l. Airspora

DR and $c_{Ha}$ were generally slightly higher in conifer monocultures than in unmanaged oak, beech and mixed stands (Figure 1), although no significant difference in spore abundance was found between these two groups of sites. The highest median densities of *H. annosum* s.l. airborne inoculum were measured in the old spruce stand by both AVST (320 spores·m$^{-3}$) and WDE (134 spores·m$^{-2}$·h$^{-1}$; Supplementary Data, Tables S1 and S2). The lowest values pertained to the oak stand (median = 9 spores·m$^{-3}$) and to the spruce stand at site 4 (median = 0 spores·m$^{-2}$·h$^{-1}$, mean = 1 spore·m$^{-2}$·h$^{-1}$).

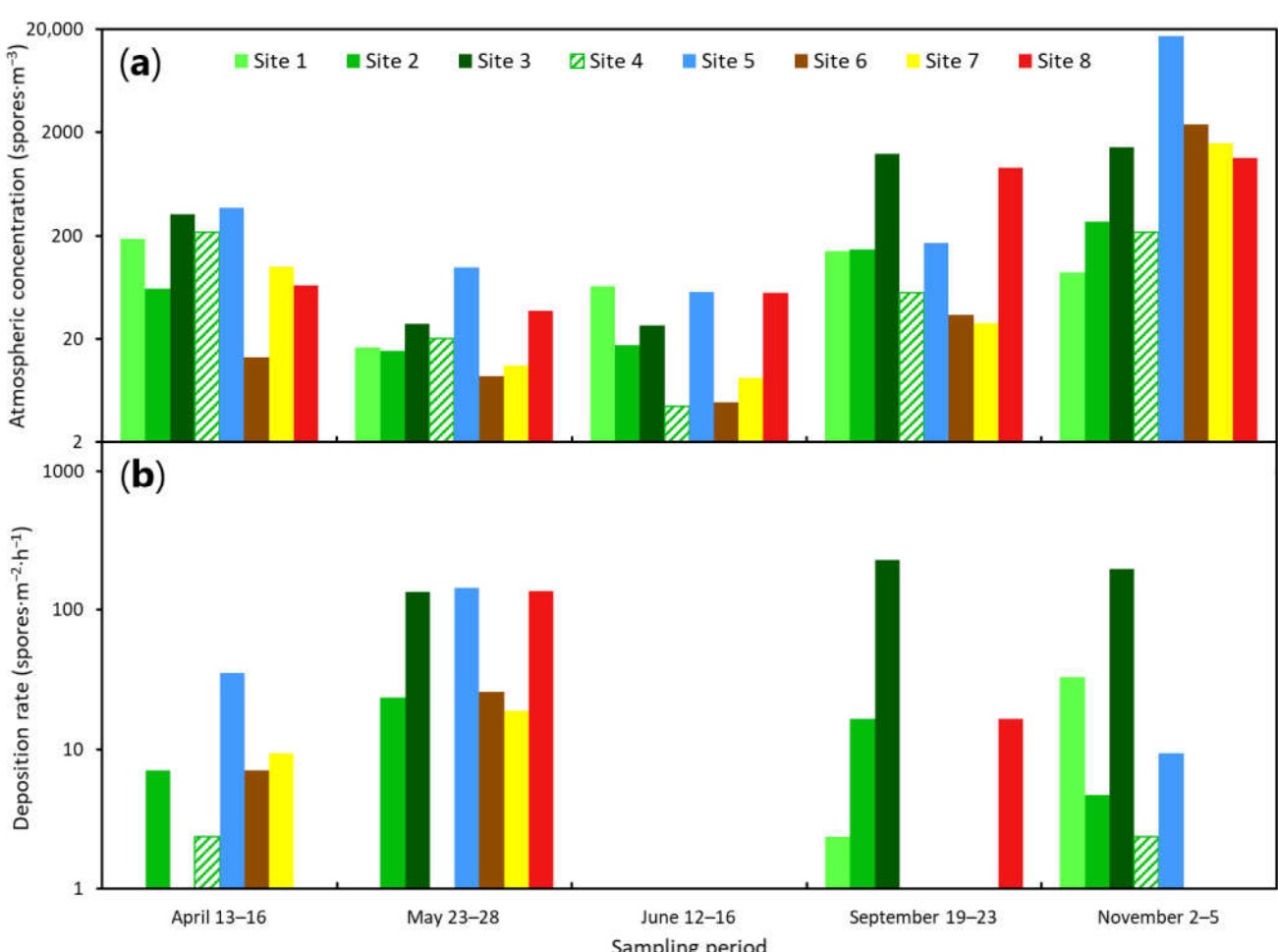

**Figure 1.** Temporal variation in spore dispersal of *H. annosum* s.l. in eight forest stands, determined by the simultaneous use of (**a**) automatic volumetric spore trap and (**b**) wood disc exposure method.

Seasonal differences were revealed in *H. annosum* s.l. spore dispersal (Figure 1). DR was significantly higher in May (median = 25 spores·m$^{-2}$·h$^{-1}$) than in June, when no spores were trapped ($z$ = 3.293, $p$ = 0.0050; Supplementary Data, Table S2). $c_{Ha}$ was significantly higher in November (median = 1290 spores·m$^{-3}$) than in May (18 spores·m$^{-3}$, $z$ = 4.085, $p$ = 0.0002) and June (22 spores·m$^{-3}$, $z$ = 3.978, $p$ = 0.0003; Supplementary Data, Table S1). The

majority of spores caught by WDE in May belong to *H. parviporum*, while *H. abietinum* was predominant in November. The composition of AVST samples varied to a smaller extent during the study period (Table 3). No significant correlations were found between meteorological factors and spore densities.

## 4. Discussion

Thanks to the excellent performance of the generic qPCR assay, quantification was possible when $c_{Ha}$ was as low as 8.7 spores·m$^{-3}$, and positive detections were made even below this level. The higher sensitivity of AVST over WDE was reflected by the two-fold difference in the detection frequency of *H. annosum* s.l. Moreover, the AVST setup was shown to give a more accurate picture of the target species composition in the air, although not always the full picture. Notably, WDE species composition was found to be biased towards *H. parviporum*, whose primary host species is Norway spruce, and *H. abietinum* also seemed to be overrepresented. On the other hand, AVST failed to detect these two species on several occasions. These inconsistencies are not easy to interpret. Brūna et al. [28] found that *H. annosum* s.s. colonized a significantly larger area than *H. parviporum* on spruce wood discs inoculated by conidial suspensions. Other studies recorded either faster growth of *H. parviporum* vs. *H. annosum* s.s. in spruce wood [29], or no difference in their spread [30]. Contradictions might be explained by the differing woody substrates analyzed or the types of inoculum used. Overall, present results support the notion that active spore traps are more efficient in capturing the airborne fungal community than static traps [31,32].

A limitation of the AVST method is its lack of ability to distinguish basidiospores from conidia. According to Hsiang et al. [33], conidia may make up around 40% of the airborne inoculum of *Heterobasidion*, although this might have been an overestimation [34]. As the formation and release of conidia strongly depend on environmental (microclimatic) factors [34], their actual abundance in the air can largely vary among forest types. Moreover, the fact that most conidia are multinucleate [33] would cause further difficulty in attempting to quantify them by qPCR. As demonstrated by artificial inoculations [35,36], conidia are also capable of colonizing conifer stumps and thus contribute to the infectious aerial spore load. The proportion of conidia caught by passive spore traps can be inferred by checking the presence of clamp connections of isolates from single-spore colonies [19,33].

The differences in spore densities between managed and unmanaged stands were unexpectedly small. Furthermore, the *Heterobasidion* species identified from colonies and air samples did not always correspond to the basidiome found in a given site. These findings might indicate spore dispersal from external sources into particular forest stands. Indeed, experimental sites 1–3 and 6–8 were located within a 100 km$^2$ forest mosaic where significant spore flow between neighboring stands, i.e., to a few hundred meters, could have happened [37]. The DR threshold of 10 spores·m$^{-2}$·h$^{-1}$, which poses a considerable risk of stump infection [38], was exceeded at least once at every site except site 4. Still, DR levels were generally low compared to those reported in the literature from similar European ecosystems [19,39]. DRs of *Heterobasidion irregulare* spores measured in the autumn in infected *Pinus resinosa* plantations in Québec [40] and Wisconsin [41] were within the range of values registered in the same season in conifer stands in present work. The above studies from North America used rotary arm spore collectors coupled with differing qPCR platforms and master mixes, which indicates that results obtained by different approaches can be comparable.

Our data on the seasonal variation in *Heterobasidion* sporulation is in agreement with previous studies in France [39] and the Italian Alps [19], where the aerial spore load was highest in the autumn and second highest in late spring. We confirm that the difference between the summer low and the autumn peak can reach two orders of magnitude. Furthermore, the $c_{Ha}$ values and their seasonal pattern measured in a Belgian mixed forest [15] were very similar to those in the mixed stand in the present study. The sharp contrast

between the relative abundance of *H. parviporum* and *H. abietinum* spores detected by WDE in May and November, coupled with the fact that the species composition of AVST samples did not show such fluctuations, may suggest that the viability of these two species on spruce wood can change over the year. It should be noted that the extremely hot and dry weather in the first half of 2018 could have affected the sporulation of the fungus, although no significant effect of the analyzed climatic factors was shown.

## 5. Conclusions

Both methods of quantifying the airspora proved to be worthwhile, and each offered certain advantages over the other. The inspection of wood discs is a rather laborious process that requires several weeks and is prone to perceptual human error. On the contrary, by analyzing AVST samples by qPCR, the exact atmospheric concentration of *H. annosum* s.l. spores can be derived within 36 h from setting up the device. The main potential of the latter method lies in its sensitivity, speed, and accuracy of detection. However, it needs higher initial investments and does not give information about the viability of the caught spores.

Ideally, the longer-term and large-scale simultaneous use of both methods would enable a reliable correspondence between DR and $c_{Ha}$. The method of choice would then depend upon individual preferences of forest managers while results could be matched to those obtained by the other method. Monitoring the spore load of the root rot pathogen in stands provides valuable data, which may locally justify the application of control measures. Furthermore, the precise estimation of the airborne infection risk by each target species may help to decide which tree species to plant after final cuts, based on their resistances to *Heterobasidion* species.

**Supplementary Materials:** The following supporting information can be downloaded at: www.mdpi.com/article/10.3390/f13122146/s1, Table S1: Absolute quantification of AVST samples; Table S2: Numbers of *Heterobasidion* colonies on wood discs 1–4.

**Author Contributions:** Conceptualization, L.B.D., M.D. and P.S.; methodology, L.B.D., M.D. and P.S.; formal analysis, L.B.D.; investigation, L.B.D., M.D. and P.S.; resources, M.D. and P.S.; data curation, L.B.D.; writing—original draft preparation, L.B.D.; writing—review and editing, L.B.D. and M.D.; visualization, L.B.D.; funding acquisition, L.B.D., M.D. and P.S. All authors have read and agreed to the published version of the manuscript.

**Funding:** This research was funded by the Specific University Research Fund of the Faculty of Forestry and Wood Technology, Mendel University in Brno, grant numbers LDF_VP_2018043 and LDF_VP_2019034. Additionally, it was supported by the European Regional Development Fund, Project "Phytophthora Research Centre", Reg. No. CZ.02.1.01/0.0/0.0/15_003/0000453, and the European Union's Horizon 2020 research and innovation program under grant agreement No. 771271, HOMED (Holistic Management of Emerging Forest Pests and Diseases).

**Data Availability Statement:** The data presented in this study are available in the article.

**Acknowledgments:** We are grateful to Martin Mullett, Zoltán Árpád Nagy, and Tomáš Májek for technical support in fieldwork, as well as to Michal Tomšovský for valuable discussions.

**Conflicts of Interest:** The authors declare no conflict of interest.

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
