# Peer review of "Volumetric Spore Traps Are a Viable Alternative Tool for Estimating Heterobasidion Infection Risk"

_forests, doi:10.3390/f13122146_

Round 1
Reviewer 1 Report
This manuscript provides information about the use of automatic volumetric spore trapping providing an accurate method to detect Heterobasidion. Experiments were done in combination to a passive method quantifying how much Heterobasidion is located in a forest stand. As expected the automatic volumetric spore trapping was more sensitive and the authors recommend this method of detection. The manuscript is well written and there are just a few suggestions listed below to improve the paper. However, one major question needs to be discussed and explained. Heterobasidion species produce asexual spores. The production of these spores is not mentioned in the introduction or discussion. Although the ability of the asexual spores to cause infection may be debated, these airborne spores are being collected by the automated spore traps and their DNA is contributing to the amount of Heterobasidion detected. This raises questions about the threshold of spores needed for stump infection. With DNA from asexual Heterobasidion spores being collected, it is providing an inaccurate analyses of the number of basidiospores collected and the spore load present that causes infection. As the authors indicate on line 93, lots of conidiophores of Heterobasidion are produced and seen on the wood discs used for passive spore detection.
Line 15 – why are they distinct?
Lines 26-30 Asexual spores are also being detected by spore traps? Information is needed on these spores in the paper since the automatic volumetric spore traps collect them as well as basidiospores.
Line 71 – why use distinct?
Line 83 – How long was exposure?
The following recent papers using an automative rotary arm spore collector for Heterobasidion detection have been published and may be useful in the discussion:
Bérubé, J.A.; Potvin, A.; Stewart, D. Importance of local and long-distance Heterobasidion irreugulare aerial basidiospore dispersal for future infection centres in thinned red pine plantation in Quebec. For. Chron. 2017, 93, 241–245.
Bérubé, J.A.; Dubé, J.; Potvin, A. Incidence of Heterobasidion irregulare aerial basidiospores at different locations in southern Quebec. Can. J. Plant Pathol. 2017, 40, 34–38.
Otto, E., B. Held, S. Redford and R. A. Blanchette. 2021. "Detecting Heterobasidion irregulare in Minnesota and Assessment of Indigenous Fungi on Pines" Forests 12, no. 1: 57. https://www.mdpi.com/1999-4907/12/1/57
Author Response
Dear reviewer,
we would like to thank you for very valuable suggestions. All were included in the current version of the manuscript.
We hope you will be satisfied.
With best regards
MD

Author Response
Dear reviewer,
thank you for your comments and ecaluation, we provide the replies to your suggestions in the attached file. We hope we have explained our point of view sufficiently.
With best regards
MD
